# Chemical Fingerprinting of Wood Sampled along a Pith-to-Bark Gradient for Individual Comparison and Provenance Identification

**Victor Deklerck [1,2,*], Cady A. Lancaster [3], Joris Van Acker [1], Edgard O. Espinoza [4], Jan Van den Bulcke [1] and Hans Beeckman [2,*]**

[1] UGent-Woodlab, Ghent University, Laboratory of Wood Technology, Department of Environment, Faculty of Bioscience Engineering, Coupure Links 653, B-9000 Ghent, Belgium; joris.vanacker@ugent.be (J.V.A.); jan.vandenbulcke@ugent.be (J.V.d.B.)

[2] Service of Wood Biology, Royal Museum for Central Africa (RMCA), Leuvensesteenweg 13, 3080 Tervuren, Belgium

[3] U.S. Forest Service International Programs Wood Identification and Screening Center 1490 East Main Street, Ashland, OR 97520, USA; cady.lancaster@usda.gov

[4] U.S. National Fish and Wildlife Forensic Laboratory, 1490 East Main Street, Ashland, OR 97520, USA; ed_espinoza@fws.gov

**\*** Correspondence: victor.deklerck@gmail.com; Tel.: +32-478-40-22-73 (V.D.); hans.beeckman@africamuseum.be (H.B.); Tel.: +32-2769-5611 (H.B.)

**Abstract:** *Background and Objectives:* The origin of traded timber is one of the main questions in the enforcement of regulations to combat the illegal timber trade. Substantial efforts are still needed to develop techniques that can determine the exact geographical provenance of timber and this is vital to counteract the destructive effects of illegal logging, ranging from economical loss to habitat destruction. The potential of chemical fingerprints from pith-to-bark growth rings for individual comparison and geographical provenance determination is explored. *Materials and Methods:* A wood sliver was sampled per growth ring from four stem disks from four individuals of *Pericopsis elata* (Democratic Republic of the Congo) and from 14 stem disks from 14 individuals of *Terminalia superba* (Côte d'Ivoire and Democratic Republic of the Congo). Chemical fingerprints were obtained by analyzing these wood slivers with Direct Analysis in Real Time Time-Of-Flight Mass Spectrometry (DART TOFMS). *Results:* Individual distinction for both species was achieved but the accuracy was dependent on the dataset size and number of individuals included. As this is still experimental, we can only speak of individual comparison and not individual distinction at this point. The prediction accuracy for the country of origin increases with increasing sample number and a random sample can be placed in the correct country. When a complete disk is removed from the training dataset, its rings (samples) are correctly attributed to the country with an accuracy ranging from 43% to 100%. Relative abundances of ions appear to contribute more to differentiation compared to frequency differences. *Conclusions:* DART TOFMS shows potential for geographical provenancing but is still experimental for individual distinction; more research is needed to make this an established method. Sampling campaigns should focus on sampling tree cores from pith-to-bark, paving the way towards a chemical fingerprint database for species provenance.

**Keywords:** chemical fingerprint; DART TOFMS; timber; geographical provenancing; *Pericopsis elata*; *Terminalia superba*

## 1. Introduction

Wood is one of the most important natural resources and has the advantage that it can be produced in a sustainable way [1]. Legal systems have been established to maximally assure conservation or sustainable logging of timber species. However, illegal timber trading is still the most profitable natural resource crime [2], and wood, being the most important traded wildlife commodity, shows the need for an increased effort to develop and improve wood identification methods for enforcement and supply chain management. This is of main concern for tropical timber, since it is estimated that 30%–90% of the volume is harvested illegally [3–5]. Countering illegal logging and trade is vital for forest conservation and sustainable logging and is only feasible by creating and implementing legal instruments. The negative effects of illegal logging range from ecological damage (for example forest degradation) to social problems and economic loss for the exporting countries [5–9]. Two main considerations for monitoring and regulating the illegal timber trade are (1) identifying which species is being traded (species identification) and (2) determining where the timber comes from (geographical provenancing).

The Convention on International Trade in Endangered Species of Wild Flora and Fauna (CITES) aims at ensuring that the international trade of specimens of plants or animals does not threaten their survival (cites.org). Recently, consumer countries have put an increased focus on the harvesting location of timber [10]. Although false claims of geographic origin are likely the most important form of illegal trade in tropical timber [5], the options to verify origin are limited. Stable isotopes and DNA analysis remain among the most promising techniques. The use of stable isotopes has shown success for different species and regions [11–14]; however, the technique is accompanied by challenges [5]. The further use and practicality of stable isotopes for timber provenancing will be determinant on the development of databases for additional taxa and regions of interest [10]. DNA analysis has shown potential in several studies, both in determining geographical provenance as well as in tracking individual logs [5,15–18]. The main obstacles here are the development of discriminating markers, limited reference databases and the methodological difficulties of extracting useful DNA out of wood [10].

Although the previously mentioned techniques have shown promise, broad application is hindered by time and cost constraint. While shipping containers await clearance in the harbor, a fast and accurate screening method is needed for the expedient processing of shipments. This is where mass spectrometry might play a role. A first indication for the potential of mass spectrometry was the discrimination of different origins of oak wood based on volatile compounds [19]. Recently, the speed and accuracy of Direct Analysis in Real Time Time-Of-Flight Mass Spectrometry (DART TOFMS) has been shown for species identification [20–22]. With DART TOFMS, a wood sliver is introduced to an ambient ionization source where the small molecules held by the heartwood (or other plant tissue) are ablated from the surface. The mass spectrum produced is in part due to the metabolome, or the small molecule profile, of the tree. The metabolite abundance within a sample is expected to be mainly determined by heritability, genetic expression and environmental factors (climate, soil, etc.).

The use of DART TOFMS is gaining momentum for identifying tropical timbers, given that it is minimally destructive for finished, high-value wood products, is a fast analysis method and no sample preparation is needed, which is essential for time-restrained enforcement policies. Moreover, it is inexpensive compared to genetic or stable isotope analysis. However, its potential for individual tracking or even individual distinction is unknown. For DART TOMFS, there are few studies demonstrating origin analysis; however, since exogenous factors influence the chemical fingerprint, the origin might be determined based on this variation. Wild agarwood was distinguished from cultivated agarwood (*Aquilaria* spp.) based on single chemical fingerprints per specimen, with inferences to a geographical region between China, Borneo, Vietnam, and Thailand [23]. DART TOFMS was also used for classifying multiple sampling sites within Bolivia for two *Cedrela* species but this proved to be difficult, likely due to the limited sample number within the proposed set-up [24]. Coastal and Cascade populations of *Pseudotsuga menziesii* (Mirb.) Franco var. *menziesii* were differentiated in western Oregon based on three chemical fingerprints corresponding to mainly

consecutive tree rings along a pith-to-bark gradient [25]. Time series analysis based on mass spectra of rings can play a potential important role in assessing the site-specific conditions during the tree's life, providing, as such, the basis for geographical provenancing [25]. Stem disks or radially oriented tree cores could be the optimal sampling strategy to start building towards fingerprints that allow the determination of the geographical provenance.

We present here a proof-of-concept study to determine whether random forest analysis with chemical fingerprints from pith-to-bark growth rings by DART TOFMS allow us to distinguish between (1) individuals within species from the African tropical moist forest and (2) geographical provenances in West and Central Africa.

## 2. Materials and Methods

Heartwood slivers (fingernail size) were cut out of growth rings, referred to as samples along a pith-to-bark gradient from stem disks (Supplementary Materials Figure S1A,B). Each stem disk was from a different individual, belonging to either *Pericopsis elata* or *Terminalia superba* (Table 1). Both species are valued timber species in the Congo Basin, and *Pericopsis elata* is even listed in the CITES Appendix II. For *Terminalia superba*, stem disks were collected in three study sites in the Democratic Republic of the Congo at the southern border of the Mayombe (two in the UNESCO Man and Biosphere Reserve of Luki, one near Tshela) and in one study site in West Côte d'Ivoire. The stem disks of *Pericopsis elata* were sampled in Biaro (Democratic Republic of the Congo). For both species, stem disks were collected in previous studies and were archived in the Tervuren Wood Collection (Royal Museum for Central Africa, Tervuren, Belgium) [26,27]. The stem disks were cross-dated, which allowed us to take wood slivers from individual rings (dated to the calendar year) as the smallest sample unit and as such account for intra-individual chemical variability. For the four *Pericopsis elata* disks, rings across multiple transects (ring sampling trajectories from pith to bark) were sampled. These transects were indicated in previous studies and they were included here to account for potential within-ring variability.

**Table 1.** Species, Tw accession number (Tervuren Wood collection) for the individual (disk), country (Democratic Republic of the Congo – DRC, Côte d'Ivoire – CDI), transect and sample number for the wood slivers.

| Species | Tw Number of the Disk | Country | Transect | Number of Tree Rings/Samples |
|---|---|---|---|---|
| *Pericopsis elata* (4 individuals) | Tw60928 | DRC | 1 | 95 |
| | | | 2 | 93 |
| | | | 3 | 95 |
| | Tw60946 | | 1 | 86 |
| | | | 2 | 82 |
| | Tw60948 | | 1 | 93 |
| | | | 2 | 93 |
| | Tw60955 | | 1 | 82 |
| | | | 2 | 101 |
| *Terminalia superba* (14 individuals) | Tw60796 | CDI | 1 | 50 |
| | Tw60799 | | 1 | 50 |
| | Tw60802 | | 1 | 20 |
| | Tw60804 | | 1 | 29 |
| | Tw60815 | | 1 | 39 |
| | Tw58841 | DRC | 1 | 25 |
| | Tw58842 | | 1 | 20 |
| | Tw58843 | | 1 | 27 |
| | Tw58844 | | 1 | 33 |
| | Tw58845 | | 1 | 20 |

| Tw61004 | 1 | 36 |
| Tw61005 | 1 | 35 |
| Tw61006 | 1 | 50 |
| Tw61009 | 1 | 32 |

All wood slivers were analyzed using a DART-SVP ion source (IonSense, Saugus, MA, USA) coupled to an AccuTOF 4G time-of-flight mass spectrometer (JEOL USA, Peabody, MA, USA) (Figure 1). The slivers were placed in the heated gas stream containing electronically excited helium atoms produced by the DART ion source. Spectra were obtained in positive ion mode, with the DART ion source parameters and mass spectrometer settings as defined in previous studies [20,21,28] over a mass range of m/z (mass-to-charge ratio) 60–1100. A mass calibration standard (poly(ethylene glycol) 600 (Ultra, Kingstown, RI, USA)) was measured between every 5th sample. TSS Unity (Shrader Software Solutions, Inc. Grosse Pointe Park, MI, USA) data reduction software was used to export the text files of the mass-calibrated, centroided mass spectra (relative abundance 0% to 100% per spectra). The text-files were exported to Microsoft Excel using a 5 mmu binning and 1% abundance threshold (see [22]). Different binning and abundance thresholds were evaluated (see also [22] for a detailed explanation of the procedure) but the prediction accuracy for individuals and geographical provenance across different settings showed little variation.

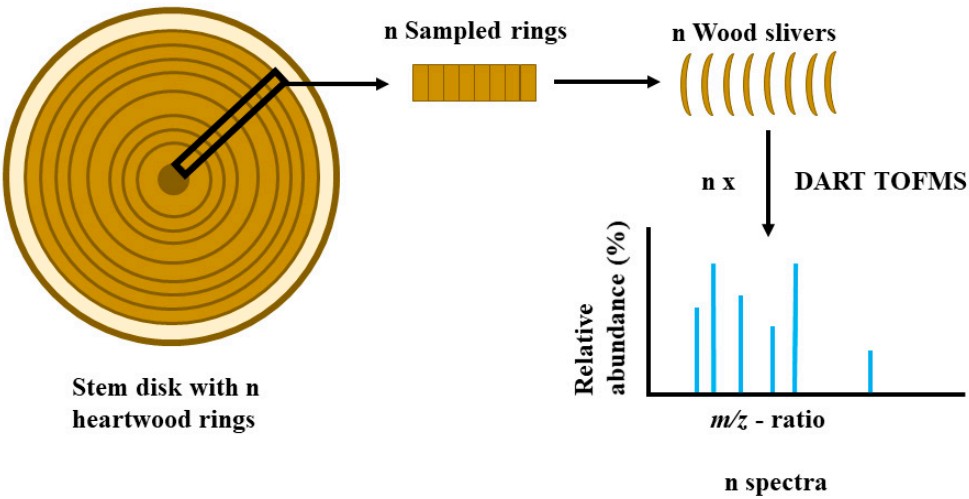

**Figure 1.** Schematic illustration of the procedure. For every sampled wood sliver, a mass spectrum is obtained.

To determine the prediction accuracies ((correctly classified samples / all samples)*100, based on cross-validation) per individual for both species, ring samples were randomly taken out of all rings per disk. For *Pericopsis elata*, all samples were pooled per disk, so no differentiation on transects was made. To determine the prediction accuracies per provenance (*Terminalia superba*, Democratic Republic of the Congo – DRC and Côte d'Ivoire – CDI), random samples were taken from sets pooled per country (see also Supplementary Materials, R-code 1). This meant that all the ring-samples from all disks were thrown together per country (2 classes: DRC and CDI), so no differentiation in individuals was made for the provenance question. This was done in an increasing sample number loop (increasing sample size for the model) for both the individual question and the provenance question separately.

For every sample number, a random forest analysis with 50 variables (i.e., ions) per split and 500 decision trees was run a 100 times, with each run having random samples but the same sample number (the sample size that is used in the random forest model). This allowed calculating an average prediction and standard deviation for a certain sample number.

A single decision tree in a random forest was based on a random sampling of the training data within the random forest. The sampling was then classified to individual or provenance (depending on

the question) based on the random choice of the 50 ions per split. This process was repeated 500 times within a random forest run (500 decision trees). This was combined with a 5-fold cross-validation, meaning that for every single random forest run (1 random sampling of the samples for a certain sample number and 500 decision trees) the data was divided into 5 equal parts, and a confusion matrix was built with 80% of the samples being in the training set and 20% of the samples being used as a test set. This confusion matrix was then built 5 times, each with a different division of the 5 parts in the training set or test set.

The final prediction accuracy and standard deviation per sample number were the average classification accuracy and standard deviation, respectively, of the 5-fold cross-validation across the 100 runs. So, the 5-fold cross-validation happened 100 times (100 random forest runs per sample number). The most important ions for the provenance classification were determined using the Gini-index and Mean Decrease in Accuracy (MDA) importance measures from random forest [25,29].

In a final test for individual disk provenance testing (whether the samples of that disk belonged to the DRC or CDI), every disk from *Terminalia superba* was removed from the dataset and used as a validation set in the random forest analysis (50 variables per split, 500 decision trees and 5 runs). The training set consisted of randomly selecting 138 samples per country, the lowest balanced dataset size possible per country, from the other disks per run. This was to avoid having samples from the same disk in both the training and prediction set when determining the country. The analysis was done in RStudio (RStudio Team, 2016) using the package randomForest [30]. The compounds important for provenance classification were, when possible, identified using the KNApSack database [31] and Mass Mountaineer Interpretation Tools software (RBC Software, Peabody, MA, USA) within a 15 mDa range, above a 2% threshold and with a stable isotope present.

## 3. Results

Prediction accuracies for the four *Pericopsis elata* individuals can be seen in Figure 2. With increasing the sample number in the model, the average prediction accuracy of individuals for the cross-validated samples increases and the standard deviation decreases. Samples from stem disk Tw60948 seem to be the easiest to identify, while it is more difficult to allocate samples from Tw60955 correctly. Clearly, the prediction accuracy increases with increasing the sample number, but flattens out from around 40 samples onward.

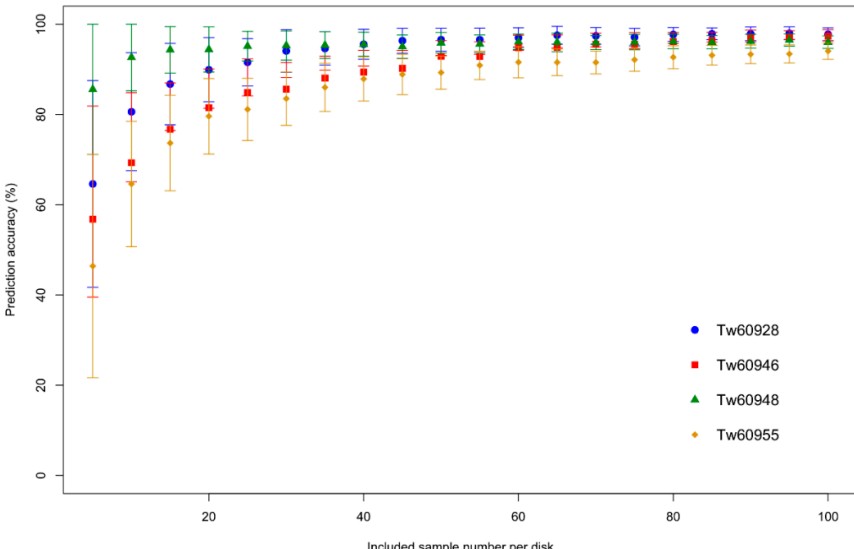

**Figure 2.** Average prediction accuracy for a random ring sample to be classified in the correct *Pericopsis elata* disk through random forest with five-fold cross-validation and increasing sample number per disk (from five to 100 samples, per five samples, with 100 runs per sample number). The bars indicate the standard deviation on the average prediction accuracy per 100 runs.

The same analysis was done for the 14 *Terminalia superba* individuals (Figure 3, see Supplementary Materials for the separate figures with standard deviations). For some individuals, the prediction accuracy does not exceed 80% (Tw61004, Tw60804, Tw61009, Tw60815), while Tw58843, for example, is almost perfect. We should note that the sample sizes were smaller compared to *Pericospsis elata*, and that for the same sample number the results look similar.

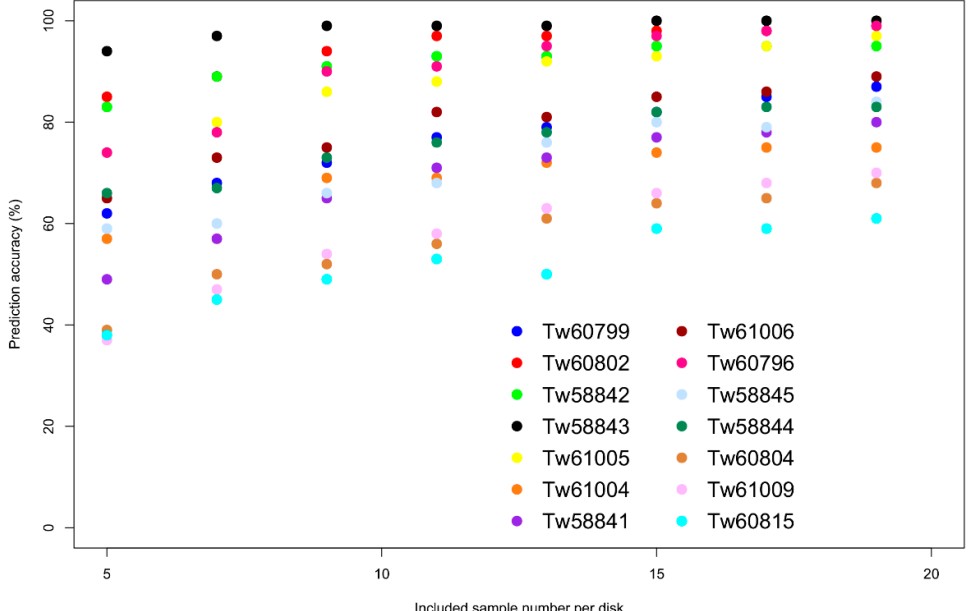

**Figure 3.** Average prediction accuracy for a random sample to be classified in the correct *Terminalia superba* disk through random forest with five-fold cross-validation and increasing sample number (from five to 19 samples, per two samples, with 100 runs per sample number). See Figure 2 in Supplementary materials for individual graphs with standard deviation bars.

Figure 4 shows the frequency (% of rings the ion is present in) of ions present in 50% or more of the tree rings for either CDI or DRC. The ions that could be identified in the KNApSack database (search term *Combretaceae*) are beta-sitosterol (*m/z* 397.38), stigmasterol (*m/z* 395.364), trimethoxyphenanthrene (*m/z* 303.123), 2,7-Dihydroxy-3,4,6-trimethoxyphenanthrene (*m/z* 302.12), quinic acid (*m/z* 193.084, *m/z* 194.085) and catechol (*m/z* 112.05). The most notable ions (biggest frequency differences between countries) are *m/z* 177.054 (present in 90.29% of the rings for DRC, present in 28.13% of the rings for CDI), *m/z* 191.072 (69.78% – DRC, 11.17% – CDI) and *m/z* 395.364, stigmasterol (3.60 % – DRC, 52.13% – CDI).

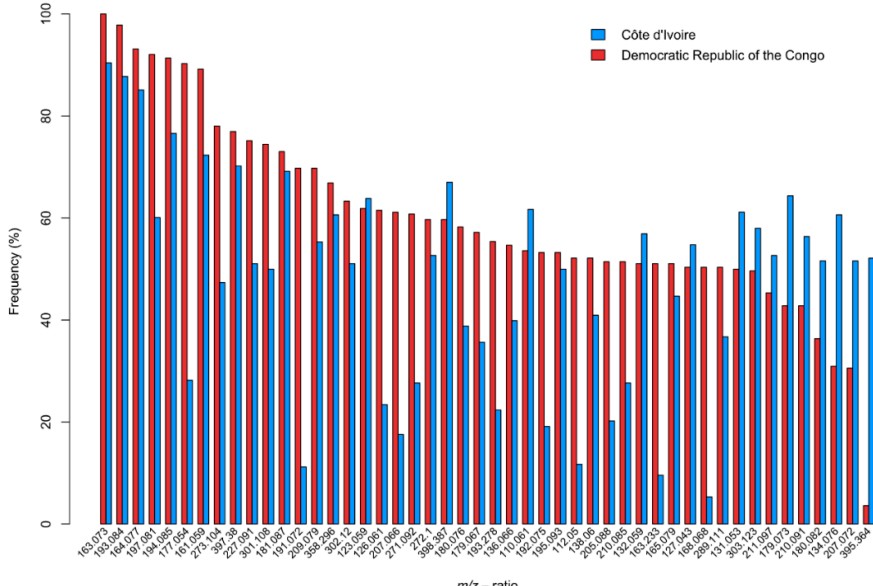

**Figure 4.** Bar plot showing all ions that are present in 50% or more of the tree rings (*Terminalia superba*) for a certain provenance ranked from highest frequency (% of rings of the ion is present in, frequency) to lowest for the DRC. The corresponding frequency for the ions in the other country when it is not present in 50% of the tree rings is shown as well. Y-axis indicates frequency (%) and the x-axis the *m/z* ratio.

Figure 5 shows the resulting prediction accuracies of the random forest cross-validation for provenance classification. Only 25 samples per country are needed to get an 80% cross-validation prediction accuracy for both CDI and DRC. For a lower number of samples, the standard deviations are high. Sample identification for the DRC appears to be more difficult compared to CDI. The most important ions to discern between provenances, based on random forest classification with 25 random samples per country (10 repeats) and the Gini-index and MDA, are: *m/z* 123.053, 134.07, 138.054, 186.075, 204.087, 251, 207.066, 277.2.

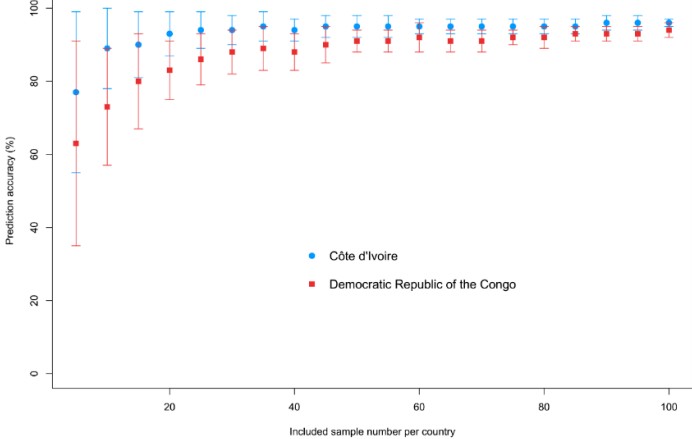

**Figure 5.** Prediction accuracy as a function of the number of samples for the provenancing of stem disks from *Terminalia superba* from Côte d'Ivoire and the Democratic Republic of the Congo through random forest analysis with five-fold cross-validation. The bars indicate the standard deviation per 100 runs.

The random forest analysis on the country level with balanced training set and each *Terminalia superba* disk used as validation can be seen in Table 2. Overall, the tree rings of the different disks are successfully placed in the correct country, although the country classification of Tw60802 (CDI) appears difficult with only 43% of the samples placed in the correct country.

**Table 2.** Country classification per disk for *Terminalia superba*. The samples of the given disk are excluded in the training set. The number of samples per disk (*n*), the average prediction success (%success) and standard deviation (%std) across the five random forest runs are given.

| Disk | Country | *n* | %Succes | %Std |
|------|---------|-----|---------|------|
| Tw60796 |  | 50 | 94.5 | 4.6 |
| Tw60799 |  | 50 | 72.0 | 3.6 |
| Tw60802 | CDI | 20 | 43.0 | 2.5 |
| Tw60804 |  | 29 | 96.6 | 3.8 |
| Tw60815 |  | 39 | 70.3 | 4.2 |
| Tw58841 |  | 25 | 100.0 | 0.0 |
| Tw58842 |  | 20 | 86.0 | 5.8 |
| Tw58843 |  | 27 | 92.6 | 4.1 |
| Tw58844 |  | 33 | 98.8 | 2.4 |
| Tw58845 | DRC | 20 | 79.0 | 8.6 |
| Tw61004 |  | 36 | 97.8 | 1.1 |
| Tw61005 |  | 35 | 100.0 | 0.0 |
| Tw61006 |  | 50 | 89.2 | 1.6 |
| Tw61009 |  | 32 | 93.1 | 1.3 |

## 4. Discussion

The determination of timber product origin is vital for regulating and stopping the illegal timber trade. To achieve this, either individual logs can be tracked throughout the supply chain, or the provenance of the timber can be determined afterwards. Log markings and documentation allow tracking of individual logs, but these are prone to falsification, especially between the concession and the mill [15]. The potential of DNA fingerprints to track single logs from concession to mill was shown with *Intsia palembanica* (merbau) in Indonesia [15] but more in depth research is needed. Although the consistent tracking of individuals throughout the supply chain would heavily obstruct fraudulent practices, it requires time and cost investment that may not be as practical or feasible to sample every individual. Also, the collection of fresh material might be done at the concession level, but when a wood product has to be traced back, the degradation of the DNA in processed wood might be a problem (see [32]). Although stable isotope analysis has shown its potential for agricultural products (see [33]), its success when using timber remains limited. *Erythrophleum* spp. (tali) was traced back to concession in Cameroon and Congo Republic based on stable isotopes and DNA microsatellites [5]. Concession assignment was successful with genetic markers, but not based on the isotopic signature, potentially due to low spatial variation in environmental conditions. Similar success with genetic methods was achieved by [16], who used DNA fingerprinting to determine whether cambium samples of *Entandrophragma cylindricum* (sapelli) originated from the forest concession in Cameroon.

Time and cost are important factors that need to be taken into account when discussing both timber identification and timber provenancing techniques. DART TOFMS allows for a rapid analysis and versatile sampling. From Figure 2, it is clear that there are chemotypical variations between individuals and that, in sufficiently small individual groups, the classification of a blind sample to the correct individual is possible, predicated by the knowledge that the blind sample comes from the small group. This indicates a homogeneity of the chemotype through time within an individual. As expected, the prediction accuracy of the random forest model increases with increasing sample number. It should be noted that for *Pericopsis elata*, only four individuals were included in the set-up compared to *Terminalia superba*. Thus, not surprisingly, the latter had lower accuracies. However, when looking at both figures (Figures 2 and 3), we can see that, within the same sample number

range, the accuracies are similar. Individual differences for *Pericopsis elata* disks have been shown based on ring-width series and stable isotopes [34]. These techniques are less straightforward to strictly separate disks or to determine whether a certain wood product comes from an individual tree. DART TOFMS is less restrictive, as it only requires a single wood sliver. As for all techniques, a database of individual trees is needed, preferably with multiple spectra per individual, for example obtained via tree cores. There is more variability in the classification accuracies for *Terminalia superba* (Figure 3). As such, individual distinction might become difficult with DART TOFMS as the number of individuals increases. Also, random forest as a classification technique might not be the best choice when the number of classes increases, and other techniques should be investigated. As such, individual distinction in the current methodology is unattainable and only comparing the individuals between each other is currently feasible.

Individual tracking will not always be possible, but determining the provenance of timber is still necessary. Separating *Terminalia superba* between the CDI and DRC is possible, and the prediction accuracy increases with increasing the sample amount. Differences in ion frequency between countries are shown in Figure 4. The above-indicated most notable ions could be a first indication of provenance. Important ions (*m/z*), based on the Gini-index and MDA, do not show important frequency differences between countries (Figure 4). This suggests that the relative abundance of an ion within a ring seems to be more important compared to its frequency throughout the rings. It is important to note that when complete disks are removed from the dataset and used as validation, the results range from 43% to 100 %, depending on which disk we try to classify to country (Table 2). It is important to note here that this eliminates the possibility that samples of one disk are in both the training and prediction set and, as such, gives a better estimation on how feasible the actual provenancing of an unknown sample is. Although most country classifications for a disk are correct, the rings of Tw60802 were misplaced most frequently, but the disk itself is clearly discerned from other disks (Figure 3). This suggests that the chemotype of Tw60802 is more unique compared to the other disks. Ring samples from this disk are easily placed together, but not in a random sampling group across disks for country identification. Of course, trees do not follow political borders, and the country delineation is an artificial one. Not every tree within a country will follow the same exact chemical signature.

The effect of site-specific characteristics and local climate on the chemical fingerprint of an individual could be the basis for this geographical provenancing, as individuals growing in a similar environment might have more similar chemotypes. Also, the consistency of the signal throughout the tree may be the foundation of understanding the chemotype differences of geographically disparate species. Species distribution plays an important role in geographical provenancing because it determines the potential exporting countries or regions, which can be important for law enforcement. The distribution of *Terminalia superba* ranges from Sierra Leone to Angola [26,35]. Although there is currently no need to regulate the trade in *Terminalia superba*, it would still be interesting to collect tree cores in future fieldwork across the distribution pattern of the species, or a wide range of tropical species for that matter, to develop country or regional chemical fingerprint databases. As trees do not follow country borders, perhaps chemical fingerprints based on the local growing environment would make more sense. Identifying climate responsive molecules should be possible by combining weather records with metabolite profiles [25]. These molecules could then be the basis for determining the geographical provenance of timber based on differences in climate between regions. It should be noted here that other factors are likely to play a role as well, for example soil, age, vitality of the tree, metabolite heritability and more. The two regions within the respective countries (DRC and CDI) where the stem disks were collected are both tropical, but differ in average annual rainfall [26]. The DRC study sites are characterized by a semi-evergreen rainforest with a five-month dry season (May to September/October) and the average annual precipitation is 1168 mm. The CDI study site has an evergreen moist rainforest, a three-month dry season (December to February) and an average annual precipitation of 1650 mm. The soils are pretty similar, with orthic ferralsols and ferric acrisols in the DRC study sites, and ferralsols and acrisols in the CDI study site.

The Global Timber Tracking Network (globaltimbertrackingnetwork.org) developed a sampling guide [36] that can be used as a basis for sampling expeditions, and that shows how to collect material for different identification or provenancing methods. A good sampling strategy for geographical provenancing with DART TOFMS would be to collect tree cores (analyze the heartwood), and a database of different samples, individuals, countries and/or climatic regions could be built. Tree cores allow for the creation of fingerprints from multiple growth rings, going from pith to bark, and could be the basis for delivering this sample amount. The number of individuals included in the test set-up was limited (four for *Pericopsis elata*, 14 for *Terminalia superba*), but avoided extended field sampling. As fieldwork and building a database takes time and money, it is important to first focus on highly targeted species and regions. Remote sensing can play a big role here, as it provides a practical approach to map changes associated with logging activities [37]. Satellite images at multiple temporal and spatial resolutions can reveal the rate and extent of deforestation, thus increasing the transparency in the forestry sector [38]. Hence, it becomes easier to monitor more remote areas, moreover, crucial areas can be pinpointed where sampling campaigns will be necessary to provide reference material to combat the oncoming illegal timber trade. A combination with different methods to determine the provenance would make a stronger case compared to a single technique [10]. As indicated, even though stable isotope ratios and DNA analysis are costly and not always straightforward, both techniques are in constant development and have shown their power as well. More recently, [39] have shown that pith-to-bark anatomical variability can indicate environmental change during a tree's life. This could strengthen the provenance determination in combination with chemical fingerprints. Also, the anatomical change could help explaining the potential variability in chemical fingerprints from pith-to-bark, this is currently understudied.

## 5. Conclusions

To enforce timber trade regulations, applicable geographical provenancing techniques are vital. Several techniques are being developed and further explored, each with their own opportunities and shortcomings. Combining DART TOFMS spectra of growth rings along a pith-to-bark gradient with random forest classification allows us to separate the country of origin (Democratic Republic of the Congo or Côte d'Ivoire) for individuals of *Terminalia superba.* The prediction accuracy increases with the training set size and is dependent on which individual is being classified to origin. Relative abundances of ions appear to contribute more to country differentiation compared to frequency differences. This study further complements work showing that mass spectral data can be used for distinguishing individuals from disparate geographic regions (see Finch et al., 2017 and Espinoza et al., 2014). This study is a proof-of-concept, to assess the potential of growth ring chemical fingerprints to distinct individuals and geographical provenances. The sample amounts are limited, and more research is needed to see how these results will translate to a broader analysis. We established that there are chemotype differences between individuals but individual tracking with DART TOFMS is still in its experimental stage and is far off from being a valid technique. Future field sampling campaigns should focus on collecting tree cores, which can be analyzed by DART TOFMS, paving the way towards the creation of a spectral database for geographical regions.

**Supplementary Materials:** The following are available online at www.mdpi.com/xxx/s1, Figure S1 A and B (supplementary for sampling scheme Materials and Methods): Sampling strategy to cut wood slivers per growth ring out of a stem disk; Figure S2 (supplementary for Figure 3): Prediction accuracies and standard deviations per individual of *Terminalia superba* (see Figure 3 main text); R-code 1 (supplementary for the data analysis and to create Figure 5): R-code of the random forest and cross-validation to determine the prediction accuracies per country. This code is similar for Figure 2 and 3.

**Author Contributions:** V.D., C.A.L., E.O.E., H.B. conceptualized this study; the methodology was developed by V.D., C.A.L., E.O.E. and J.V.d.B.; model construction by V.D. and J.V.d.B.; validation was done by J.V.A., H.B. and E.O.E.; formal analysis was done by V.D., C.A.L. and J.V.d.B. The original draft was prepared by V.D. and C.A.L. Review and editing was done by all authors and the visual optimization by V.D., C.A.L. and J.V.A., supervision was done by J.V.A., E.O.E., J.V.d.B. and H.B. Project administration was done by V.D. and H.B.

Funding acquisition was done by J.V.A., E.O.E., J.V.d.B. and H.B. All authors have read and agreed to the published version of the manuscript.

**Funding:** This research was funded by the HERBAXYLAREDD BELSPO Brain program (Belgian Science Policy – Federaal Wetenschapsbeleid) of the Belgian Federal Government (BR/143/A3/HERBAXYLAREDD).

**Acknowledgments:** The authors would like to thank Stijn Willem (UGent – Woodlab), Pam McClure and Erin Price (US Fish and Wildlife Forensic Laboratory) for their help with the sample preparation. The authors would also like to thank Thomas Mortier for his advice on the coding aspect (UGent - Department of Data Analysis and Mathematical Modelling). The findings and conclusions in the article are those of the authors and do not necessarily represent the views of the U.S. Fish and Wildlife Service.

**Conflicts of Interest:** The authors declare no conflict of interest.

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
