# Peer review of "Chemical Fingerprinting of Wood Sampled along a Pith-to-Bark Gradient for Individual Comparison and Provenance Identification"

_forests, doi:10.3390/f11010107_

Round 1
Reviewer 1 Report
This is an interesting and well presented paper, on a topic of importance. The chemical fingerprinting process appears novel, and the authors describe clearly recognized limitations in current methodology with resulting variation and uncertainty. This is however a meaningful contribution from which additional research can develop.
From an editorial perspective, I suggest that success and std values presented in Table 2 be to x.x%, not x.xx%, unless the authors can demonstrate that measurement precision actually supports x.xx% level of significant figures.
As a suggestion toward future research to validate the DART TOFMS chemical fingerprinting process, I would recommend benchmark consideration of several different North American tree species where from anatomical characteristics the wood is just about impossible to separate. Examples include northern and southern red oak, and the broader mix of different species making up red and white oaks, the different species of southern pine, and coastal and interior grown Douglas-fir.
Reviewer 2 Report
This manuscript is claimed as a proof-of-concept study to investigate the accuracy of using random forest to determine species and provenance of timber/logs based on chemotype fingerprints measured with DART-TOFMS. Tree rings from trees of two valuable species, Pericopsis elata and Terminalia superba, from two counties (DRC and CDI) were used as study samples.
DART-TOFMS is gaining attention in many fields as it is fast, does not require samples preparation and can be done in atmospheric environment. It is specially useful in combating illegal logging and timber trading as it has high potential for quick wood species identification and provenancing. The manuscript has originality and importance to be published in the journal.
The Introduction section is well written and covers the relevant references. Materials and methods section could be improved to give clearer description of sampling scheme and data analysis process. The presentation of the results is somewhat not very easy to follow. Maybe I am not familiar wit the subject, but it would be more informative if the author could explain for example the definition of the most notable ions (Line 200) - is it because these ions have the most difference in frequency between two countries? The Conclusion section needs to be re-written -- it is more like a summary, not conclusions.
a minor comment: the repetition of result (lines 305-307) is not necessary, reword it.
